# Key mechanistic features of the trade-off between antibody escape and host cell binding in the SARS-CoV-2 Omicron variant spike proteins

Weiwei Li[1,2,8], Zepeng Xu[1,3,8], Tianhui Niu [4,8], Yufeng Xie [1,5], Zhennan Zhao[1], Dedong Li[1], Qingwen He[1], Wenqiao Sun[1], Kaiyuan Shi[1], Wenjing Guo[1,2], Zhen Chang [6], Kefang Liu[1], Zheng Fan [7✉], Jianxun Qi [1,2✉] & George F Gao [1✉]

## Abstract

Since SARS-CoV-2 Omicron variant emerged, it is constantly evolving into multiple sub-variants, including BF.7, BQ.1, BQ.1.1, XBB, XBB.1.5 and the recently emerged BA.2.86 and JN.1. Receptor binding and immune evasion are recognized as two major drivers for evolution of the receptor binding domain (RBD) of the SARS-CoV-2 spike (S) protein. However, the underlying mechanism of interplay between two factors remains incompletely understood. Herein, we determined the structures of human ACE2 complexed with BF.7, BQ.1, BQ.1.1, XBB and XBB.1.5 RBDs. Based on the ACE2/RBD structures of these sub-variants and a comparison with the known complex structures, we found that R346T substitution in the RBD enhanced ACE2 binding upon an interaction with the residue R493, but not Q493, via a mechanism involving long-range conformation changes. Furthermore, we found that R493Q and F486V exert a balanced impact, through which immune evasion capability was somewhat compromised to achieve an optimal receptor binding. We propose a "two-steps-forward and one-step-backward" model to describe such a compromise between receptor binding affinity and immune evasion during RBD evolution of Omicron sub-variants.

**Keywords** Omicron; Immune Evasion; Receptor Binding; SARS-CoV-2; Sub-variants
**Subject Categories** Microbiology, Virology & Host Pathogen Interaction; Structural Biology

## Introduction

Since the announcement on November 26, 2021, the fifth SARS-CoV-2 variants of concern (VOC), Omicron (B.1.1.529), has rapidly disseminated worldwide (Xu et al, 2022b). Concerningly, the virus keeps evolving to adapt to the host and sub-variants have been rapidly emerging one after another, including BF.7, BQ.1, BQ.1.1, XBB, XBB.1.5, BA.2.86 and JN.1 (Han et al, 2022; Kurhade et al, 2023; Uraki et al, 2023). Despite our great effort to develop vaccines and monoclonal antibodies (MAbs) for herd immunity establishment, the newly emerged sub-variants keep adapting to current immune background and demonstrated further immune evasion (Addetia et al, 2023; Cao et al, 2023, 2022b; He et al, 2023; Huang et al, 2022; Yue et al, 2023).

Receptor binding and immune escape have been recognized as two major drivers for the evolution of receptor-binding domain (RBD) of the viral spike (S) protein, which is responsible for receptor recognition (Ma et al, 2023; Telenti et al, 2022; Wang et al, 2020). Studies found that occurrence of new RBD mutations are correlated with both receptor binding and immune evasion (Telenti et al, 2022), and recent quantification studies indicated that the latter is more significantly correlated with mutation occurrence (He et al, 2023; Huang et al, 2022; Ma et al, 2023). The interaction between two factors has attracted much attention. Studies revealed that affinity-enhancing mutation N501Y is epistatic to immune evasive mutations, indicating intertwined interactions between the two driving forces (Moulana et al, 2022; Starr et al, 2022). Cao et al further incorporated ACE2 binding ability, codon usage constraint and the immune escape profiles to estimate the evolution trend of RBD (Cao et al, 2023). However, the mechanism of the interplay between the two factors, although explored, remains elusive.

Since the emergence of VOCs, we and others have been constantly monitoring the receptor binding characteristics of the VOCs, esp. the predominant Omicron sub-variants (Addetia et al, 2023; Cao et al, 2023, 2022a; Han et al, 2022, 2021; Zhao et al, 2023). It has been noticed that R493Q reversion significantly

[1]CAS Key Laboratory of Pathogen Microbiology and Immunology, Institute of Microbiology, Chinese Academy of Sciences (CAS), Beijing, China. [2]University of Chinese Academy of Sciences, Beijing, China. [3]Faculty of Health Sciences, University of Macau, Macau SAR, China. [4]Air Force Medical University, Air Force Medical center, PLA, Beijing, China. [5]Department of Basic Medical Sciences, School of Medicine, Tsinghua University, Beijing, China. [6]Department of Pathogen Microbiology, School of Basic Medical Sciences, Capital Medical University, Beijing, China. [7]Institutional Core Facility, Institute of Microbiology, Chinese Academy of Sciences (CAS), Beijing, China. [8]These authors contributed equally: Weiwei Li, Zepeng Xu, Tianhui Niu. ✉E-mail: fanzh@im.ac.cn; jxqi@im.ac.cn; gaof@im.ac.cn

strengthened the receptor binding, but might be deleterious for immune escape (Cao et al, 2023; Zhao et al, 2023). Based on these results, we hypothesize that the R493Q reversion is a compromise between receptor binding and immune evasion.

In this study, we evaluated the binding properties of BF.7, BQ.1, BQ.1.1, XBB and XBB.1.5 RBDs to human ACE2 (hACE2) and determined the cryo-EM structures of hACE2 in complex with BF.7, BQ.1, BQ.1.1, XBB and XBB.1.5 RBDs, respectively. From the structures, we found that it is R493 but not Q493 that is regulated by R346T substitution through long-range conformation alterations. Furthermore, we found the mutations in the RBD from BA.2 to BA.4/5 appear as a compromise between the receptor binding and immune evasion. These results enhance our comprehension of the relationship between receptor binding and immune evasion of these Omicron sub-variants.

# Results

## Omicron BF.7, XBB, XBB.1.5, BQ.1 and BQ.1.1 RBD exhibit comparable receptor binding strength within the optimal affinity scope

Since May 2022, the BA.5.2 sub-variant has grown rapidly. Subsequently, the population infected by BF.7, BQ.1 and BQ.1.1 rapidly increased. Since November 2022, XBB and XBB.1.5 have emerged, and XBB.1.5 subsequently became the predominant pandemic strain worldwide. To evaluate the entry efficiency of these emerging Omicron sub-variants, we prepared the VSV-backbone pseudoviruses harboring the S proteins from the SARS-CoV-2 prototype (PT) or Omicron sub-variants including BA.1, BA.2, BA.2.75, BA.4/5, BQ.1, BQ.1.1, BF.7, XBB and XBB.1.5 to infect Vero cells (Fig. 1A). Pseudovirus entry assay indicated that all Omicron sub-variants have enhanced entry capacities compared to PT (Fig. 1A). Notably, BF.7, XBB and XBB.1.5 have a higher entry efficiency than BA.5 sub-variant (Fig. 1A), which was previously the dominant strain worldwide.

The phylogenetic analysis showed that BF.7 could evolve from BA.5 and gained an extra R346T substitution (Fig. 1B). The R346 site is a mutation hotspot, and previous studies showed that it plays pivotal roles in immune escape and receptor binding (Cao et al, 2023; He et al, 2023; Li et al, 2022). XBB is a recombinant between BM.1.1.1 and BJ.1 sub-variants (Tamura et al, 2023), and XBB.1.5 is a descendant of XBB, which has more substitutions, including G252V and S486P in S (Fig. 1B). In addition, BQ.1 is another descendant of BA.5 sub-variant, which has gained K444T, N460K mutations compared to BA.5, and has evolved into BQ.1.1 and BQ.1.1.18 by gaining R346T substitution (Fig. 1B).

Receptor binding is a key step for SARS-CoV-2 infection. To evaluate the influence of these emerging mutations on the receptor binding capacity, surface plasmon resonance (SPR) was performed using the soluble proteins of both S proteins and RBDs of these sub-variants, including BA.5, BQ.1, BQ.1.1, BF.7, XBB and XBB.1.5. BQ.1 ($K_D = 19.07 \pm 4.08$ nM), BQ.1.1 ($K_D = 14.70 \pm 0.37$ nM) and BF.7 ($K_D = 13.37 \pm 0.29$ nM) S proteins to hACE2 are comparable to that of BA.4/5 ($K_D = 14.37 \pm 0.41$ nM). Notably, compared to the XBB S ($K_D = 123.7 \pm 6.18$ nM), the XBB.1.5 S ($K_D = 25.97 \pm 0.94$ nM) has an increased receptor binding affinity to hACE2 (Fig. 1C, Table EV1). Importantly, the changing trends of RBD-hACE2 affinities and

S-ACE2 affinities of these sub-variants were found to be consistent (Fig. EV1, Table EV1).

## The structures of Omicron BF.7, XBB, XBB.1.5, BQ.1 and BQ. 1.1 RBDs in complex with hACE2

To unveil the molecular mechanisms of Omicron BQ.1, BQ.1.1, XBB, XBB.1.5 and BF.7 RBD/hACE2 complexes, the cryo-EM structures of these complexes were determined at resolutions of 2.71, 2.69, 2.80, 2.91 and 2.47 Å, respectively (Appendix Figs. S1–S5, Table EV2). Moreover, the crystal structures of BF.7 and BQ.1.1 RBD binding to hACE2 were also solved at resolutions of 3.46 and 3.40 Å, respectively, for detailed molecular interactions (Fig. EV2, Table EV3).

As previously reported, the binding interface of SARS-CoV-2 RBD in complex with ACE2 was divided into two patches (Liu et al, 2021b; Tang et al, 2022; Wang et al, 2020; Xu et al, 2022a). The residues participating in H-bonds and salt bridges contacts in the RBD of the five complex structures are all conserved as expected (Table EV4).

In patch 1, Y453 and N487 of RBD form hydrogen bonds (H-bonds) with Q24 and H34 in the hACE2, respectively (Fig. 2A–E). In addition, RBD N487 also forms an H-bond with hACE2 Y83 in the BQ.1.1 RBD/hACE2, XBB RBD/hACE2 and XBB.1.5 RBD/hACE2 complex structures (Fig. 2B–D). Q493 of RBD binds to K31 of hACE2 in the four out of five complexes except for XBB RBD/hACE2 complex, in which it was S490 of RBD but not Q493 binding to hACE2 K31 (Fig. 2C). In the BQ.1.1 RBD/hACE2, XBB RBD/hACE2 and XBB.1.5 RBD/hACE2 complexes, RBD N477 contacts to hACE2 S19 with an H-bond (Fig. 2B–D). In all the five cryo-EM complex structures, hACE2 H34 possesses two structural conformations. In the XBB RBD/hACE2 and XBB.1.5 RBD/hACE2 complex structures, hACE2 H34 forms an additional H-bond with RBD S494 (Fig. 2C,D). The main chain of BQ.1.1 RBD F490 forms an H-bond with hACE2 K31 that is not observed in other four complexes (Fig. 2B).

The H-bonds and salt bridges interaction network in patch 2 of all these five cryo-EM complex structures are extremely similar. Y449, R498, T500 and G502 of RBD form H-bonds or salt bridges with D38, Y41, Q42 and K353 of hACE2 in the four out of five complexes except for BQ.1 RBD/hACE2 complex, in which hACE2 Q42 does not form H-bonds with the corresponding residue in the RBD (Fig. 2A–E).

## The role of hotspot mutation residue 486 for receptor binding

Compared with XBB RBD, XBB.1.5 RBD only possesses an S486P substitution, but the binding affinity of XBB.1.5 RBD to hACE2 is ~2.7 folds higher than that of XBB RBD (Fig. 3E). The sequence alignment indicated that the RBD residue 486 is a hotspot mutation (Fig. EV3). The F486 of PT RBD is substituted by serine (S), valine (V) or proline (P) in some of the SARS-CoV-2 sub-variants (Fig. EV3). We aligned the complex structures that possess different 486 residues in the RBD based on the α1 and α2 helixes of hACE2 (Fig. 3A–D). The side chain of F486, V486 and P486 point to the α1 and α2 helixes of hACE2 (Fig. 3B–D). On the contrary, the side chain of S486 points in the opposite direction (Fig. 3A). As previously reported (Wang et al, 2020), F486 inserts into a hydrophobic pocket formed by F28, L79, M82 and Y83 in the hACE2 (Fig. 3C). Though the side chain of V486 is shorter than

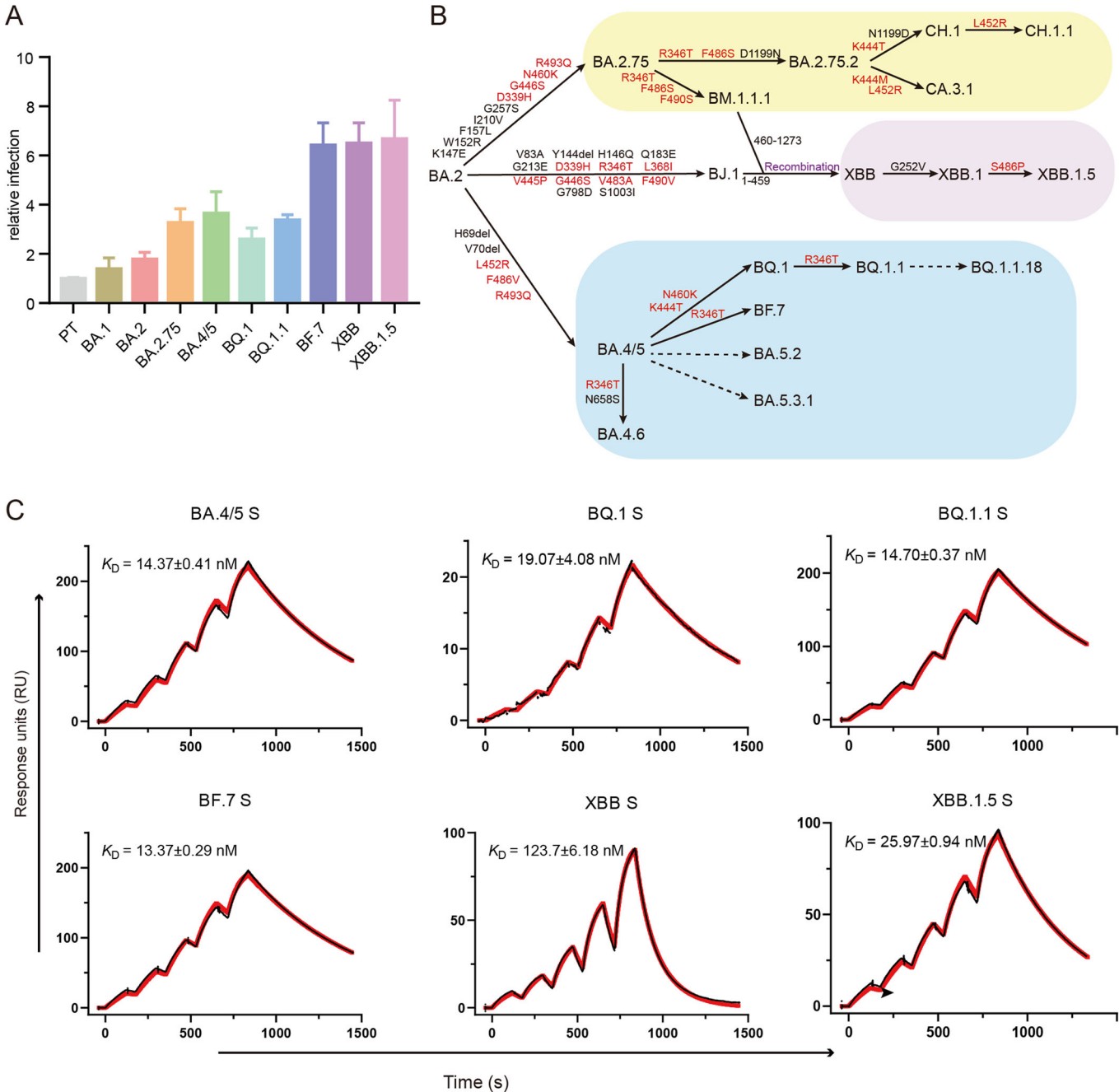

**Figure 1. Receptor binding characteristics of SARS-CoV-2 Omicron sub-variants.**

(A) Pseudovirus entry assay for the PT and several Omicron sub-variants, including BQ.1, BQ.1.1, XBB, XBB.1.5 and BF.7. Infectivity for each Omicron sub-variant was normalized based on the PT, and the mean of relative infection was shown as the y-axis. The error bars indicate the standard deviations (SD) for three independent experiments. It was conducted at least twice with six duplicates each time. (B) A schematic of SARS-CoV-2 Omicron sub-variants evolution and changes on S proteins. Mutations on the RBD are marked in red, others in black. (C) The SPR curves for the BA.4/5, BQ.1, BQ.1.1, BF.7, XBB and XBB.1.5 S binding to hACE2. Raw and fitted curves are represented by black and red lines, respectively. Dissociation constant ($K_D$) indicates mean ± SD from three independent repeats. Source data are available online for this figure.

that of F486, V486 is also a hydrophobic amino acid that can interact with the hydrophobic amino acids, L79 and M82, in the hACE2 through van der Waals force (VDW) (Fig. 3D). P486 interacts with M82 through VDW (Fig. 3B). However, the hydrophilic amino acid S486 cannot interact with any residues in the hACE2 (Fig. 3A).

**It is R493 but not Q493 that is regulated by residue 346 in the RBD**

Previously, we found that R346K mutation in BA.1.1 RBD majorly affects the interaction network in the BA.1.1 RBD/hACE2 interface through long-range alterations and contributes to the

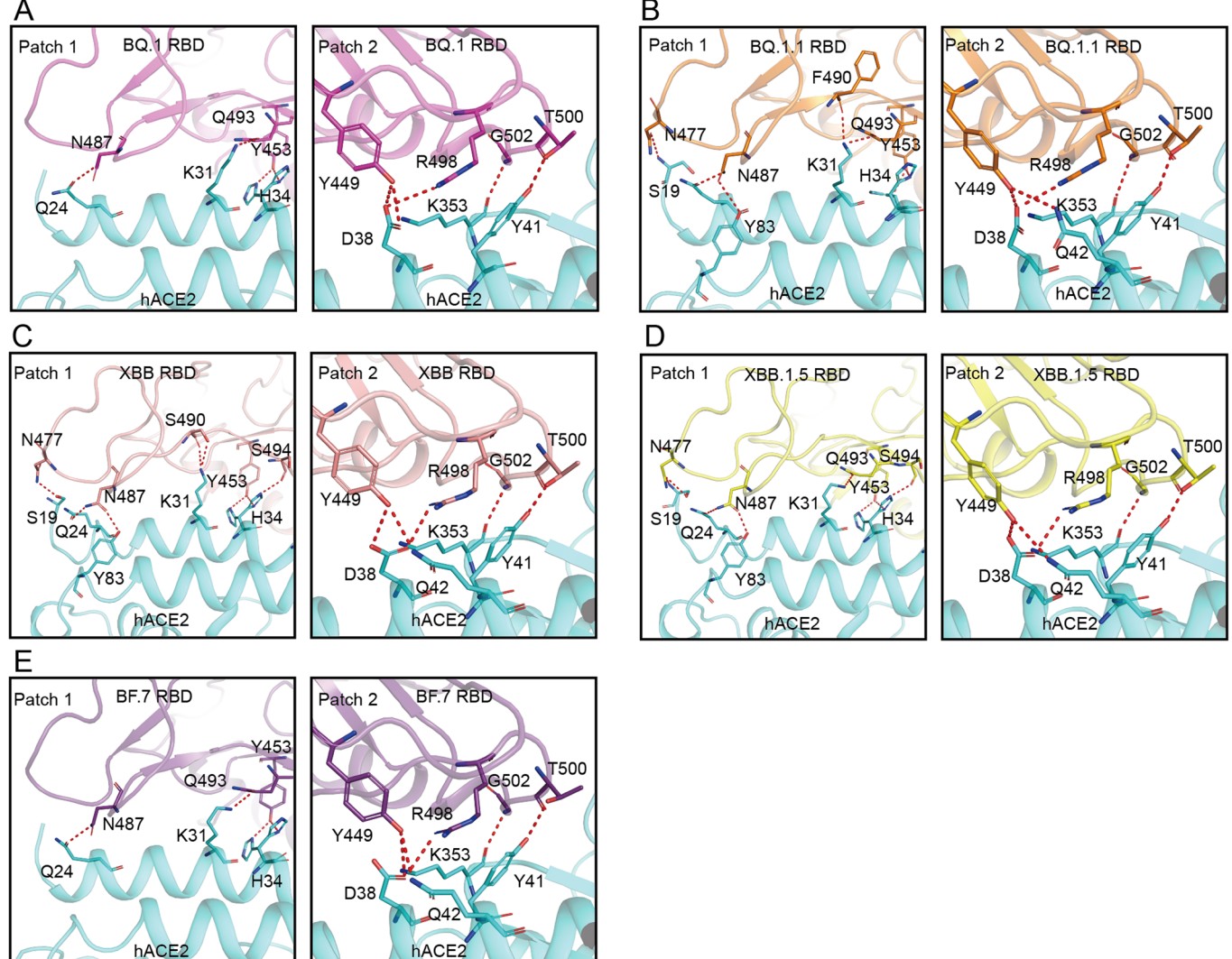

**Figure 2.** The interaction network of BQ.1, BQ.1.1, XBB, XBB.1.5 and BF.7 RBDs bound to hACE2.

(A–E) The structures of BQ.1 (**A**), BQ.1.1 (**B**), XBB (**C**), XBB.1.5 (**D**) and BF.7 (**E**) RBDs in complex with hACE2 are show as cartoons. The residues forming H-bonds or salt bridges are shown as sticks. H-bonds and salt bridges are show as red dashed lines.

higher hACE2 affinity of the BA.1.1 RBD than the BA.1 RBD (Li et al, 2022). However, it is unclear which residue on the binding interface synergizes with residue 346. In this study, we found BF.7 RBD only has R346T mutation compared with BA.4/5 RBD (Fig. 4A). But unlike BA.1.1 RBD, this substitution does not increase the binding affinity between BF.7 RBD and hACE2. Further analysis found R493 in BA.1 RBD and BA.1.1 RBD were substituted by Q493 in BA.4/5 RBD and BF.7 RBD. We hypothesized that it is R493 but not Q493 that was regulated by R346T substitution. To confirm our hypothesis, we mutated Q493 of BA.4/5 and BF.7 RBD to R493 and tested the binding affinities between these RBDs and hACE2. As we expected, the binding affinity of BA.4/5 RBD Q493R to hACE2 decreases ~6.0-fold compared with wild-type (WT) BA.4/5 RBD (Fig. 4B,C, Table EV1). On the contrary, BF.7 RBD Q493R increases ~2.3-fold affinity to hACE2 than that of WT BF.7 RBD (Fig. 4B,C, Table EV1). It means that there is only one different residue, R346T,

between BA.4/5 RBD Q493R or BF.7 RBD Q493R, but the binding affinity between them to hACE2 is of ~16.0-fold difference (Fig. 4B,C, Table EV1). The RBD of BM.1.1.1 has three substitutions (R346T, F486S and F490S) compared to BA.2.75 RBD. The binding affinity of BM.1.1.1 RBD to hACE2 ($K_D = 71.70 \pm 1.71$ nM) is lower than that of BA.2.75 ($K_D = 8.21 \pm 0.59$ nM), due to the F486S substitution. When Q493 was substituted with R493, the binding affinity of BA.2.75 RBD Q493R to hACE2 decreased ($K_D = 31.47 \pm 0.26$ nM), while the binding affinity of BM.1.1.1 RBD Q493R increased ($K_D = 28.37 \pm 0.39$ nM). These changes are mainly due to the difference in residue 346 between the BM.1.1.1 (T346) and BA.2.75 (R346) RBDs (Fig. 4B,C, Table EV1).

The structural conformation of R493 has large difference between BA.1 RBD and BA.1.1 RBD when binding to hACE2 (Fig. EV4). R493 in BA.1 RBD (R346) forms a salt bridge with E35 of hACE2. In BA.1.1 RBD (K346), R493 forms an additional H-bond with H34 and an additional salt bridge with

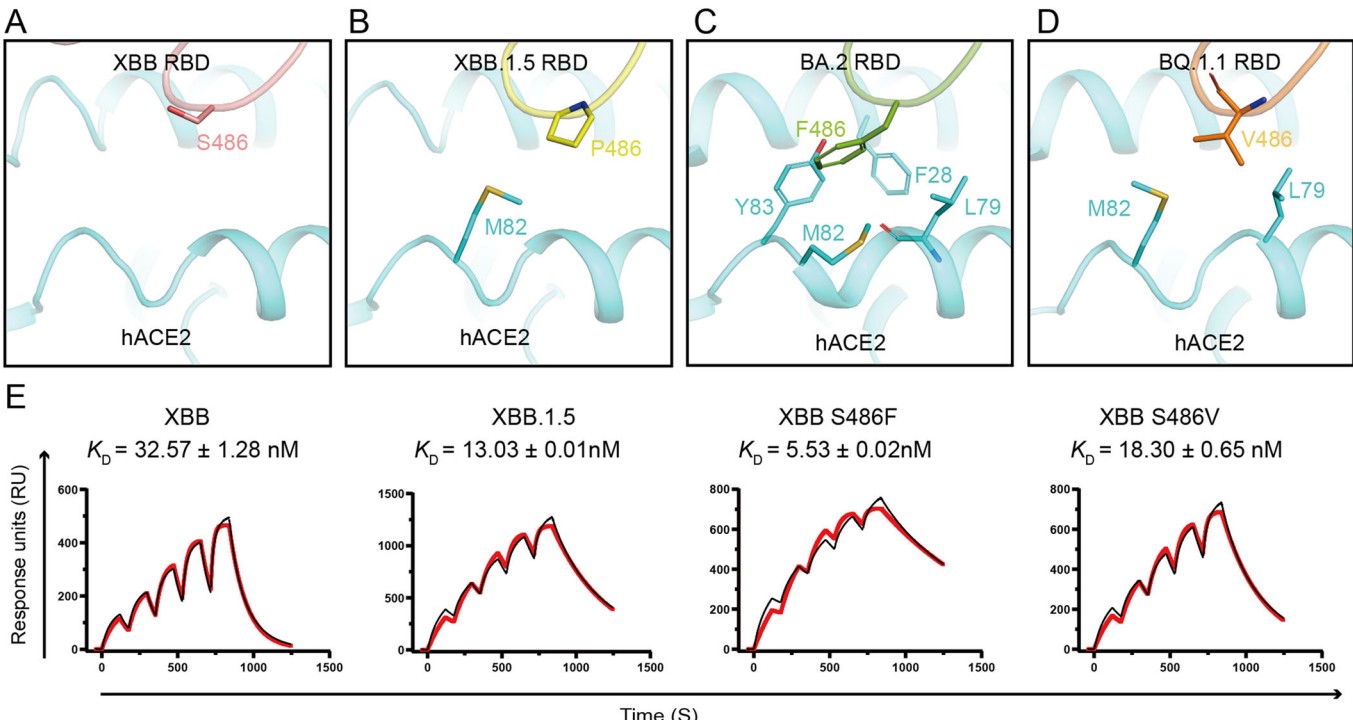

**Figure 3.  Residue 486 of RBD modulates hACE2 binding.**

(A–D) The complex structures of XBB (A), XBB.1.5 (B), BA.2 (C) and BQ.1.1 (D) RBDs bound to hACE2 are shown as cartoon. The different 486 residues and its contact residues in the hACE2 are shown as sticks. (E) The SPR analysis of RBDs with different 486 residues binding to hACE2. Dissociation constant ($K_D$) indicates mean ± SD of three independent replicates. Actual and fitted curves are colored in black and red, respectively. Source data are available online for this figure.

D38, which makes it a much higher binding affinity with hACE2 (Fig. EV4). For the BA.4/5 and BF.7 RBD, Q493 shows similar conformation in the two complexes and the interaction networks on the interface are also similar (Fig. 4D–F). So that the binding affinities of BA.4/5 and BF.7 RBD to hACE2 are similar. In the BA.4/5 RBD/hACE2 complex, Q493 forms neither H-bond nor salt bridge with hACE2 (Fig. 4G). Y449 and R498 of BA.4/5 RBD contact with D38 and Q42 of hACE2 through a strong interaction network. But when we made a Q493R mutation in the BA.4/5 RBD, R493 forms two salt bridges with D38 (Fig. 4H). But the interaction network formed by BA.4/5 RBD Y449 and R498, and hACE2 D38 and Q42 is broken, so the binding affinity to hACE2 decreased (Fig. 4H).

**The compromise between receptor binding and immune evasion**

From BA.2 RBD to BA.4/5 RBD, R493 was reversely substituted to Q493 (the same as in PT) and was fixed in subsequently emergent sub-variants. The residue on site 493 is important in both receptor binding and immune evasion, especially in escaping MAbs belonging to RBD-1 (He et al, 2023; Huang et al, 2022; Li et al, 2022; Zhao et al, 2023). To explore the effect of R493Q on receptor binding and immune evasion, as well as its role in RBD evolution, we constructed the point mutants of BA.2 RBD by substituting the distinct residues between BA.2 RBD and BA.4/5 RBD, namely F486V and R493Q (L452R is not included as this residue locates away from the interface of hACE2 and RBD-1 MAbs) and

measured the binding affinity of BA.2 RBD mutants to hACE2. The results showed that R493Q significantly increased the binding affinity with hACE2, whereas F486V sabotaged hACE2 binding (Fig. 5A). Then, as sites 486 and 493 mainly affect evasion from MAbs belonging to RBD-1 (Hastie et al, 2021), we tested the binding affinities of BA.2 RBD R493Q with representative RBD-1 MAbs screened from people infected with PT SARS-CoV-2 (BD604, BD629, P2C-1F11 and S2K146) which still exhibit RBD binding for BA.2 and BA.4/5 (He et al, 2023). We found R493Q reverse mutation strengthened binding with 3 of the 4 MAbs, whereas F486V significantly promoted the immune evasion (Fig. 5A, Table EV4).

To further analyze the interplay between receptor binding and immune evasion lying under the two substitutions, we analyzed the receptor binding between the RBDs of SARS-CoV-2 variants and hACE2, and found that the affinities to hACE2 of these variants are limited within a scope of one to two digital numbers ($K_D$ ~ 5–40 nM, normalized to results of our laboratory for comparability) (Fig. 5B). To evaluate whether antibodies induced by sub-variants prior to Omicron BA.1 restore some ability to neutralize the SARS-CoV-2 Omicron sub-variants after the Q493 reverse mutation, we used the abovementioned RBD-1 MAbs (BD604, BD629, P2C-1F11 and S2K146) to test their binding affinities to BA.2, BA.2 F486V, BA.2 R493Q and BA.4/5 RBDs (He et al, 2023; Huang et al, 2022; Li et al, 2022). Then we summarized the hACE2 binding affinity and immune evasion from RBD-1 MAbs. We found that R493Q in fact rescued the receptor binding sabotaged by F486V at the cost of its own immune evasion function. Thus, the combined effect of the two substitutions presents as compromised

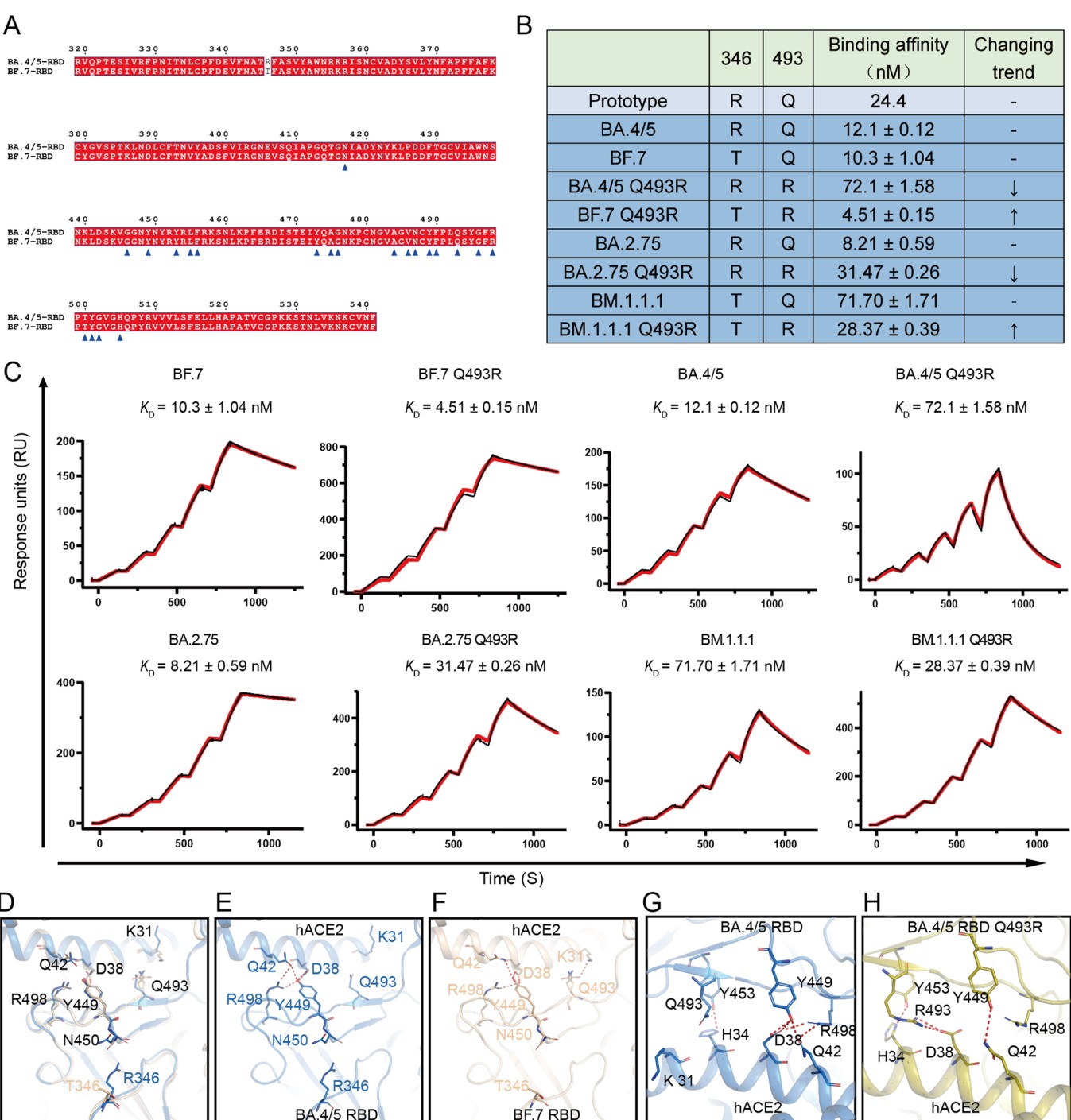

**Figure 4. The synergetic relationship between residue 346 and residue 493.**

(A) Sequence alignment of BA.4/5 RBD and BF.7 RBD. Residues directly involved in hACE2 interaction are labelled with blue triangles. Conserved residues are highlighted in red. (B) The statistics of binding affinities. The residue 346 and the counterpart residue 493 in different RBDs are included. (C) The SPR analysis of BA.4/5 RBD, BF.7 RBD and their Q493R mutants bound to hACE2. Dissociation constant ($K_D$) indicates mean ± SD of three independent replicates. Actual and fitted curves are colored in black and red, respectively. (D–F) Structure alignment of BA.4/5 RBD/hACE2 and BF.7 RBD/hACE2 complexes. (G, H) The structural comparison of BA.4/5 RBD and its mutant BA.4/5 RBD Q493R bound to hACE2. The backbone of the structures are shown as cartoon and the key residues are shown as sticks. The H-bonds and salt bridges are shown as red dash lines. Source data are available online for this figure.

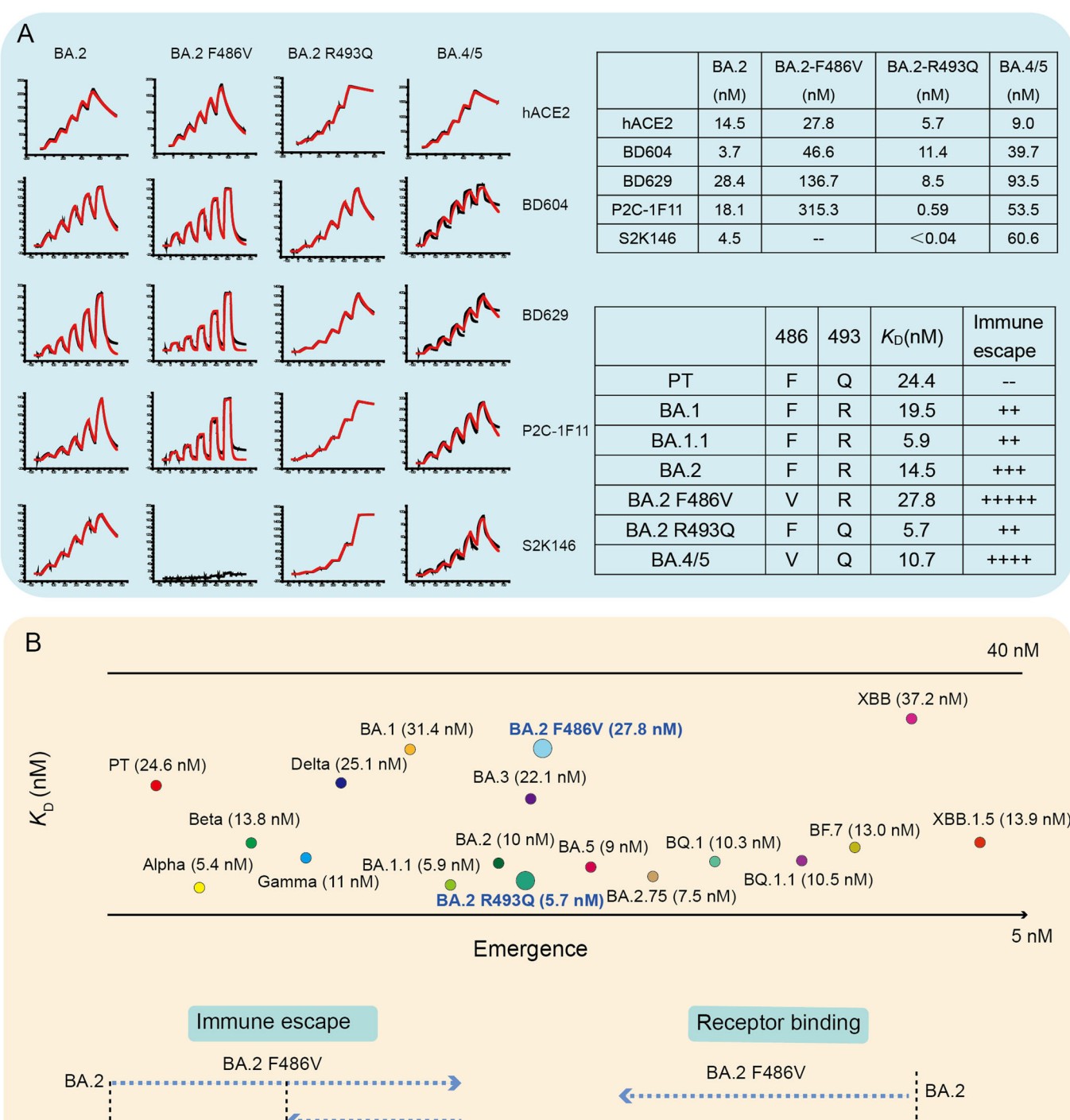

**Figure 5. The compromise between receptor binding and immune evasion.**

(**A**) The SPR curves for the binding affinities of BA.2, BA.2 F486V, BA.2 R493Q and BA.4/5 RBDs with hACE2 and representative RBD-1 MAbs (BD604, BD629, P2C-1F11 and S2K146) (left) and summary of the hACE2 binding affinity and immune evasion from RBD-1 MAbs in this and our previous work (right). Raw and fitted curves of SPR assays are represented by red and black lines, respectively. "--" mains no immune escape and "+" represents immune escape. (**B**) The summary of receptor binding affinities between RBDs of SARS-CoV-2 variants and hACE2, and the "two-steps-forward and one-step-backward" immune escape and optimal receptor binding affinity models. Source data are available online for this figure.

(strengthened yet less than BA.2 F486V) immune evasion and maintained binding affinity (Fig. 5A, Table EV4). Thus, R493Q and F486V substitutions in the RBD from BA.2 to BA.4/5 appear as a compromise between enhancing the receptor binding and sabotaging immune evasion.

Based on these results, we herein propose a "two-steps-forward and one-step-backward" model for RBD evolution (Fig. 5B). The RBD evolves to achieve further immune evasion, but is constrained by receptor binding. When the immune evasion goes too far and the binding affinity exceeds optimal scope, there will be a compromise and a substitution will occur that rescues the receptor binding but somewhat sabotage immune evasion (Fig. 5B).

## Discussion

Relationship between receptor binding and immune evasion is an important scientific question to understand SARS-CoV-2 RBD evolution and continuously emerging variants and sub-variants. This study is designated to resolve such an important scientific question. Our data indicate that R493Q reversion is a compromise between receptor binding and immune evasion. Based on the analysis of the interplay between R493Q and F486V, we can hypothesize that immune evasion is constrained by receptor binding, and RBD will take "two-steps-forward and one-step-backward", when the binding affinity falls out of the optimal scope. Combined with the existence of substitutions solely promoting immune evasion (K356T for instance) (Cao et al, 2023), the integrated result would be further immune evasion while the binding affinity maintained within the optimal scope.

The optimal affinity scope contains an upper limit (5 nM) and lower limit (40 nM). The lower limit is easily understood as receptor binding is the prerequisite for virus entry, but why is there an upper limit? In this study, we found that BF.7 Q493R RBD containing 346T/493R exhibited unexpectedly high binding affinity with hACE2 and is higher than 346T/493Q. Previous studies also reported that R346T is an important substitution for immune evasion (Cao et al, 2023). Thus, theoretically 346T/493R is favorable for both receptor binding and immune evasion. Looking along the evolution from BA.2 (346R/493R) to BA.4/5 (346R/493Q) to XBB.1.5 (346T/493Q), there is an interesting question: why R346T did not occur in BA.2-subsequent lineages and achieve a 346T/493R pair, but instead went through two mutations (R493Q and F486V) for a similar status? Although virus carrying this 346T/493R pair has been observed in real world (13,027 sequences in GISAID as of Aug. 25th, 2023), such lineage never gained dominance either before or after Omicron VOC emergence. Is there other unidentified constraint to prevent such "optimal" combination? Can we hypothesize that there is a biologically relevant ceiling for receptor binding, above which the higher affinity, on the contrary, is detrimental for virus transmission? This question is worth further investigating, as it might help for ultimate goal to eradicate the SARS-CoV-2 virus by a live attenuated vaccine to compete to any exiting or emerging SARS-CoV-2.

There are several limitations to this study. Although receptor binding is a crucial step for SARS-CoV-2 infection, it is important to acknowledge that there are numerous other factors that can influence virus infectivity. The abundance of S copies on the surface of the

authentic or pseudoviruses may vary among sub-variants due to mutations, potentially altering avidity, rather affinity, for both receptors and antibodies. However, directly testing these changes through experiments can be challenging. In addition, it is worth noting that our findings may not fully capture the impact of RBD open/closed conformations on affinity, as both S and RBD were immobilized proteins in our experimental setup. This limitation is inherent to the SPR methodology used. Furthermore, mutations occurring outside the RBD region may also affect ACE2 binding by influencing RBD exposure in the "up" conformation, which requires further investigation.

## Methods

### Cells

HEK293F suspension-cultured cells (Gibco, Cat#11625-019) were cultured at 37 °C in SMM 293-TII Expression Medium (Sino Biological, Cat# M293TII), and BHK-21 adherent cells (ATCC CCL-10) were cultured at 37 °C in Dulbecco's modified Eagle medium (DMEM) supplemented with 10% fetal bovine serum (FBS).

### Gene cloning

As previously reported (He et al, 2023; Huang et al, 2022; Liu et al, 2021a; Zhao et al, 2023), the coding sequences of RBD of SARS-CoV-2 (residue 319–541), RBDs of variants (residue 319–541) and hACE2 (residues 19–615, GenBank:NP_001358344) including the Hexa-His tag sequence at the C-terminus were inserted into the pCAGGS vector. The variable region of S304, BD604, BD629, P2C-1F11, and S2K146 MAbs fused with the constant region of immunoglobulin G1, was synthesized and cloned into pCAGGS vectors.

### Protein expression and purification

HEK293F suspension-cultured cells (Gibco, Cat#11625-019) were cultured at 37 °C in SMM 293-TII Expression Medium (Sino Biological, Cat# M293TII). The pCAGGS-S304, pCAGGS-hACE2, and pCAGGS-RBDs were expressed in HEK293F cells. The cells were transfected with a mixture of 0.5 mg plasmid and 1.5 mg polyethyleneimine at a density of $2 \times 10^6$ cells/mL in 500 mL. A supplement (Sino Biological, Cat# M293-SUPI) was added to the culture system 24 h and 72 h after transfection.

Cell culture supernatants were collected after a 5-day infection. The hACE2 and RBD proteins were purified using His-Trap HP columns (GE Healthcare) and the HiLoad™ 16/600 Superdex™ 200 pg column (GE Healthcare). The MAbs were purified using a Protein A affinity column (GE Healthcare). MAbs were generated via papain digestion and further purified using a Protein A column (GE Healthcare) and gel filtration with a HiLoad™ 16/600 Superdex™ 200 pg column (GE Healthcare). Purified proteins were stored in a buffer containing 20 mM Tris-HCl and 150 mM NaCl (pH 8.0).

The hACE2 protein produced by HEK293F cells was used for Cryo-EM and SPR assay. The RBD proteins produced by HEK293F cells were used for Cryo-EM and SPR. The S304 fab protein produced by HEK293F cells was used for Cryo-EM. The proteins for SPR assay were stored in PBST buffer (1.8 mM $KH_2PO_4$, 10 mM $Na_2HPO_4$ (pH 7.4), 137 mM NaCl, 2.7 mM KCl, and 0.05% (v/v) Tween 20).

## SPR assay

The SARS-CoV-2 PT and variants S and RBD proteins were transferred into PBST buffer (1.8 mM $KH_2PO_4$, 10 mM $Na_2HPO_4$ (pH 7.4), 137 mM NaCl, 2.7 mM KCl, and 0.05% (v/v) Tween 20) and immobilized on CM5 chip. Serially diluted hACE2s were then flowed over the chip in PBST buffer. Binding affinities were measured using a BIAcore 8K (GE Healthcare) at 25 °C in the single-cycle mode. To test the binding affinities of MAbs with different RBDs, MAbs immobilized on CM5 chip and RBDs flowed over the chip in PBST buffer. Binding kinetics were analyzed with BIAcore Insight software (GE Healthcare) using a 1:1 binding model. Gradient concentrations of hACE2 from 200 nM to 12.5 nM with two-fold dilution flowed over the chip in PBST buffer. The CM5 chip (GE Healthcare) was regenerated using 10 mM Glycine-HCl (pH 1.5). $K_D$ values of SPR experiments were obtained with BIAcore 8K Evaluation Software (GE Healthcare), using a 1:1 binding model. The values indicate the mean ± SD of three independent experiments.

## Protein complex preparation

Purified hACE2 was mixed and incubated with BF.7 RBD, XBB RBD, XBB.1.5 RBD, BQ.1 RBD and BQ.1.1 RBD at a 1:1.5 molar ratio (hACE2 to RBD) on ice for about 2 h. The six mixtures were then purified on Superdex™ 200 10/300 GL column (GE Healthcare) in a buffer containing 20 mM Tris (pH 8.0) and 150 mM NaCl. The XBB.1.5 RBD/hACE2 was mixed with S304 Fab at a 1:1.5 molar ratio (hACE2-RBD to Fab) on ice for about 2 h and purified on Superdex™ 200 10/300 GL column (GE Healthcare) with 20 mM Tris (pH 8.0) and 150 mM NaCl. Purified complex proteins (BF.7 RBD/hACE2, XBB RBD/hACE2, XBB.1.5 RBD/hACE2/S304 Fab, BQ.1 RBD/hACE2 and BQ.1.1 RBD/hACE2) were concentrated to 2 mg/mL for cryo-EM using a 30-kDa cut-off Ultracon concentrator (Millipore).

## Cryo-EM sample preparation and data acquisition

For the BF.7 RBD/hACE2, BQ.1 RBD/hACE2, BQ.1.1 RBD/hACE2, XBB RBD/hACE2 and XBB.1.5 RBD/hACE2/S304 complexes, droplets (3.5 μL) of the complex at approximately 0.2 mg/mL was frozen with the graphene oxide (GO) grids (GO on Quantifoil Au R1.2/1.3, 400 mesh), using a Vitrobot Mark IV (Thermo Fisher Scientific) with blotting for 2 s. The datasets were collected using a 300 kV Titan Krios transmission electron microscope equipped with a Gatan K3 detector and GIF Quantum energy filter. Movies were collected in super-resolution counting mode at pixel size of 0.425 Å. The exposures were performed with a dose rate of 15 $e^-$/pixel/s and an accumulative dose of 50 $e^-$/Å². The defocus ranges of these datasets were $-1.0 \sim -2.0$ μm.

## Image processing and 3D reconstruction

The drift of all stacks was motion corrected with MotionCor2 (Zheng et al, 2017). All the micrographs were subjected to cryoSPARC (Punjani et al, 2017). The contrast transfer function (CTF) parameters were estimated using patch CTF estimation (Sanchez-Garcia et al, 2021). To avoid the preference orientation problem in RBD/hACE2 datasets, we trained three Topaz models to potentially pick particles in specific orientations using BQ.1 RBD/hACE2 dataset. In detail, iterative 2D classification of 195,736 particles picked from 214 micrographs separate three uneven groups of particles. The major

"C"-like group contains 22,806 particles while minor sickle-like or mushroom-like groups only contains 5815 particles and 1252 particles, respectively. These particles were subjected to Topaz training procedure respectively and the union of the particles picked by models could potentially engage more orientations.

For the BQ.1 RBD/hACE2 complex, a total of 183,262 particles were picked from 214 micrographs by applying the aforementioned Topaz models. One round of heterogeneous refinement separated a dominant class containing a subset of 101,822 best particles, showing the clear features of secondary structural elements. The particles were subjected to non-uniform refinement, which yielded a map at 2.71 Å resolution.

The dataset of BF.7 RBD/hACE2 complex was processed similarly. Briefly, a total of 881,011 particles in 798 micrographs were picked for the following heterogeneous refinement. A dominant class (260,759 particles) was selected and used to calculate the density map at 2.47 Å resolution by non-uniform refinement and CTF refinement.

For the BQ.1.1 RBD/hACE2 complex, a total of 577,961 particles in 375 micrographs were picked. 2D classification separated a clean dataset with 510,029 particles. Two rounds of heterogeneous refinement separated a dominant class containing a subset of 151,069 best particles. The particles were subjected to non-uniform refinement and CTF refinement, yielding a map at 2.69 Å resolution.

For the XBB RBD/hACE2 complex, a total of 343.631 particles were picked for the following 2D classification and heterogeneous refinement. A dominant class (124,538 particles) was selected and used to calculate the density map at 2.80 Å resolution by non-uniform refinement.

For the XBB.1.5 RBD/hACE2/S304 complex, the three pretrained Topaz models were no longer applicable. We retrained Topaz model by 12,781 particles in several orientations, which represented the clean subset of the 120,533 particles picked in 184 micrographs subset. The Topaz model extracted 1,680,199 particles in the entire dataset of 6597 micrographs. 2D classification separated a clean dataset with 596,461 particles. Subsequently, heterogeneous refinement separated a dominant class containing a subset of 244,183 best particles. The particles were subjected to non-uniform refinement, yielding a map at 2.91 Å resolution.

## Crystallization

The sitting-drop method was used to obtain the high resolution crystals. In detail, purified complex proteins were concentrated to 5 and 10 mg/mL. Then, 0.8 μL protein was mixed with 0.8 μL reservoir solution. The resulting solution was sealed and equilibrated against 100 μL of reservoir solution at 18 °C and 4 °C. High-resolution hACE2/BQ.1.1 RBD complex crystals were grown in 0.1 M Sodium citrate 5.5, 15% w/v PEG 6000, and hACE2/BF.7 RBD grew in 0.2 M Sodium chloride, 0.1 M Tris 8.0, 20% w/v PEG 6000.

## Model building and structure refinement

The structure of RBD/hACE2 complex (PDB 6LZG) was docked into the cryo-EM density maps of the BQ.1 RBD/hACE2, BF.7 RBD/hACE2, BQ.1.1 RBD/hACE2, XBB RBD/hACE2 and XBB.1.5 RBD/hACE2/S304 complexes using Chimera v.1.14 (Pettersen et al, 2004), while the S304 region extracted from the RBD/S2E12/S309/S304 complex (PDB: 7R6X) was rigid-body docked into the map of XBB.1.5

RBD/hACE2/S304 complex. The models were manually mutated and refined iteratively in COOT and PHENIX (Adams et al, 2010; Emsley and Cowtan, 2004). The stereochemical quality of each model was evaluated using MolProbity. Structural figures were generated using PyMOL, Chimera and ChimeraX (Goddard et al, 2018).

### Production and quantification of pseudoviruses

To obtain SARS-CoV-2 PT and variants pseudoviruses, we constructed replication-deficient vesicular stomatitis virus vector backbone (VSV-ΔG-GFP) expressing the corresponding spike proteins. 30 mg of spike protein expression plasmids were transfected into HEK293T cells each 10 cm culture dish. After 24 h, The VSV-ΔG-GFP pseudoviruses were added to the transfected cell supernatant. After incubation for 2 h at 37 °C, inoculum was replaced with fresh DMEM containing both 10% FBS and anti-VSV-G antibody produced by I1HybridomaATCC®-CRL2700™. The pseudoviruses were obtained 30 h post-infection. After being filtered by 0.45 mm filters (Millipore, Cat#SLHP033RB), the pseudoviruses were aliquoted and stored at −80 °C.

0.5 U/mL BaseMuncher endonuclease (Abcam) was used to remove unpackaged RNA at 37 °C for 1 h. Viral RNA was extracted using an RNA extraction kit (Bioer Technology) and quantified by quantitative RT-PCR.

### Pseudovirus infection assays

The numbers of the pseudovirus particles for SARS-CoV-2 and its variants were normalized by the VSV L gene qPCR. Extract RNA from pseudovirus suspension and conduct qRT-PCR experiments using TAKARA's one-step reverse transcription kit (Nie et al, 2020). Calculate the relative ploidy relationship of pseudovirus particle numbers for different variants based on CT values. We diluted to have the same number of viral particles per microliter via quantitative RT-PCR. VSV specific forward primer (VSV-F): 5′-TGATACAGTACAATTATT TTGGGGAC-3′, reverse primer (VSV-R): 5′-GAGACTTTCTGT TACGGGATCTGG-3′ and VSV-probe: FAM-ATGATGCATGAT CCWGC-TAMRA. Then, 100 μL of pseudovirus was added to each well of 96-well plates containing Vero cells. After 15 h, each whole well was scanned using a CQ1 confocal image cytometer (Yokogawa) and the total numbers of GFP-positive cells were determined using the software bundled with the instrument (Yokogawa). Each group included six biological replicates, and the analysis was repeated 3–4 times. Statistical analysis was performed using Graphpad Prism 8.

### Quantification and statistical analysis

#### Binding affinity analysis

$K_D$ values of SPR experiments were obtained with BIAcore 8K Evaluation Software (GE Healthcare), using a 1:1 binding model. The values indicate the mean ± SD of three independent experiments.

## Data availability

The crystal structures of Omicron BF.7 RBD/hACE2 complex (PDB: 8WDS, rcsb.org/structure/unreleased/8WDS) and Omicron BQ.1.1 RBD/hACE2 complex (PDB: 8WDR, rcsb.org/structure/unreleased/8WDR) as well as the cryo-EM structures of Omicron BF.7 RBD/hACE2 (PDB: 8WE1/EMD-37470, rcsb.org/structure/unreleased/8WE1), Omicron BQ.1 RBD/hACE2 (PDB: 8WDZ/

EMD-37468, rcsb.org/structure/unreleased/8WDZ), Omicron BQ.1.1 RBD/hACE2 (PDB: 8WDY/EMD-37467, rcsb.org/structure/unreleased/8WDY), Omicron XBB RBD/hACE2 (PDB: 8WE0/EMD-37469, rcsb.org/structure/unreleased/8WE0) and Omicron XBB.1.5 RBD/hACE2/S304 (PDB: 8WE4/EMD-37471, rcsb.org/structure/unreleased/8WE4) complexes have been deposited in the Protein Data Bank (www.rcsb.org). This study did not generate custom computer code. Any additional information required to reanalyze the data reported in this work is available from the Lead Contact upon request.

## Peer review information

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

## Acknowledgements

We acknowledge the staff of beamline BL02U1 and BL10U2 at the Shanghai Synchrotron Radiation Facility for assistance during data collection. We thank YQ Mi, H Feng, and XX Bai at the Cryo-EM Center, Shanxi Academy of Advanced Research and Innovation for their technical support on the Cryo-EM. This work was supported by the National Key R&D Program of China (2021YFC2301401, 2020YFA0907102, 2020YFA0509202), the National Natural Science Foundation of China (92169208 to JQ), and Special Program of China National Tobacco Corporation (110202102034 to JQ).

## Author contributions

**Weiwei Li**: Resources; Formal analysis; Methodology. **Zepeng Xu**: Software; Methodology; Writing—original draft. **Tianhui Niu**: Software; Methodology. **Yufeng Xie**: Software; Methodology. **Zhennan Zhao**: Software; Methodology. **Dedong Li**: Methodology. **Qingwen He**: Methodology. **Wenqiao Sun**: Methodology. **Kaiyuan Shi**: Methodology. **Wenjing Guo**: Methodology. **Zhen Chang**: Methodology. **Kefang Liu**: Conceptualization; Formal analysis; Writing—original draft. **Zheng Fan**: Software; Supervision; Methodology. **Jianxun Qi**: Resources; Software; Supervision; Funding acquisition; Writing—review and editing. **George F Gao**: Conceptualization; Data curation; Supervision; Writing—review and editing.

## Disclosure and competing interests statement

George Gao is an editorial advisory board member at The EMBO Journal. This has no bearing on the editorial consideration of this article for publication.

# Expanded View Figures

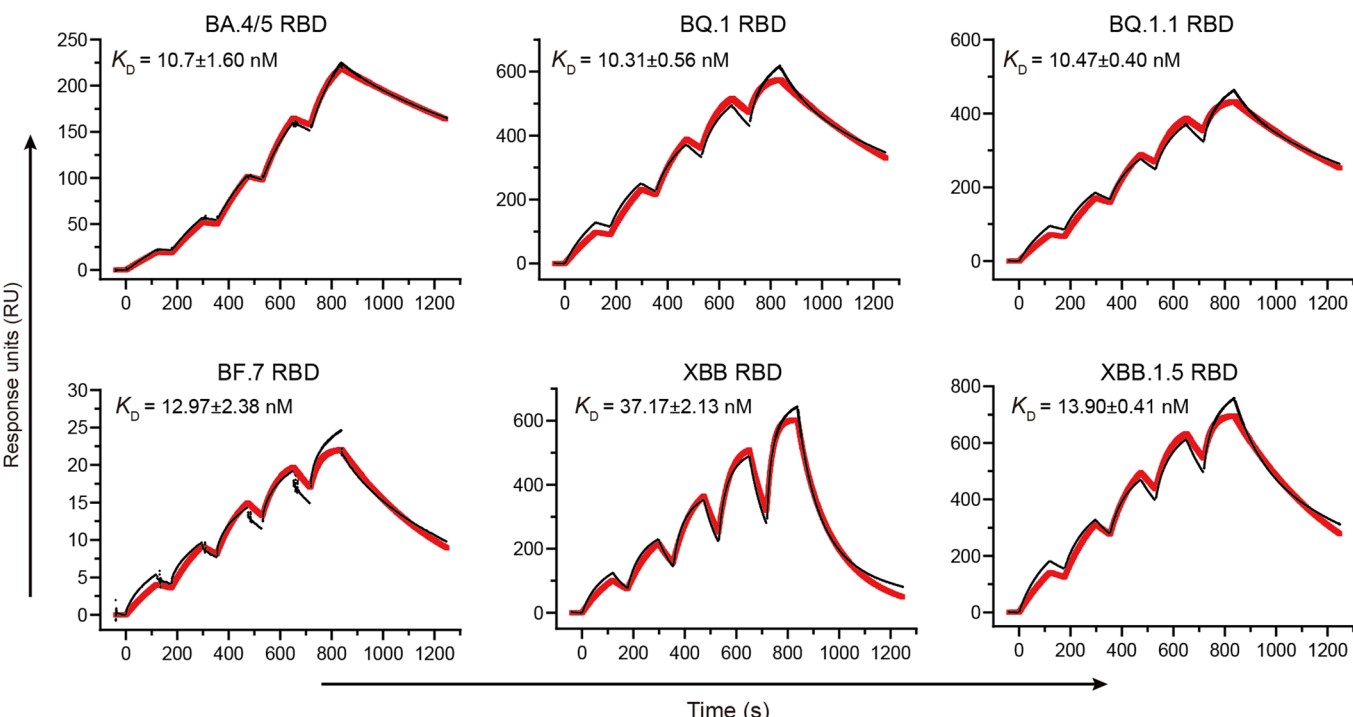

**Figure EV1.  The SPR curves for the BA.4/5, BQ.1, BQ.1.1, BF.7, XBB, and XBB.1.5 RBD binding to hACE2.**

Raw and fitted curves are represented by black and red lines, respectively. Dissociation constant ($K_D$) indicates mean ± SD from three independent repeats. Source data are available online for this figure.

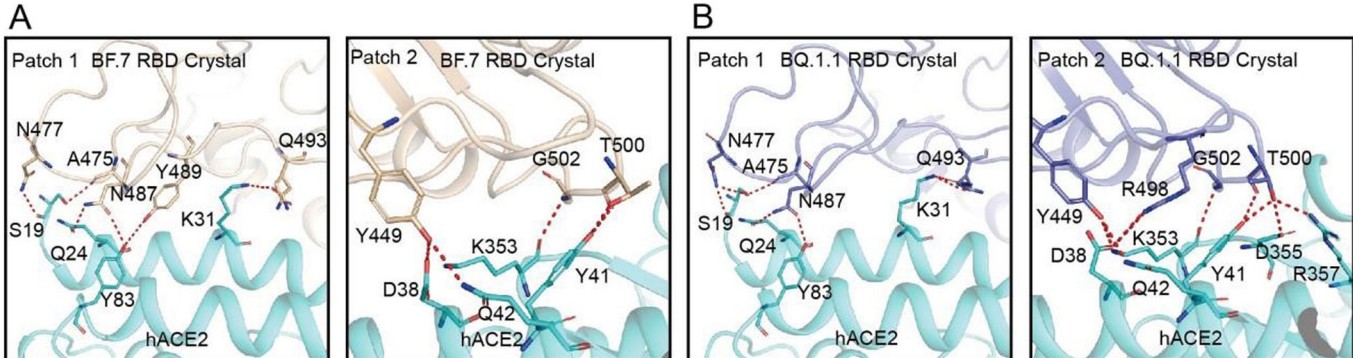

**Figure EV2. The crystal structure of BQ.1.1 and BF.7 RBDs in complex with hACE2.**

(A, B) The structures of BF.7 (A) and BQ.1.1 (B) RBDs in complex with hACE2 are show as cartoons. The residues form H-bonds or salt bridges are shown as sticks. H-bonds and salt bridges are show as red dash lines.

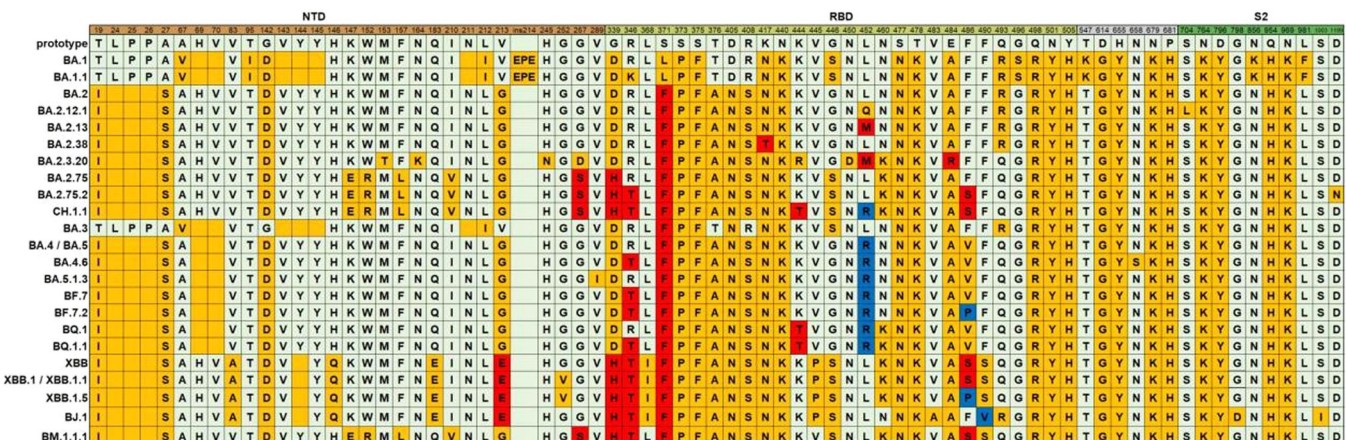

**Figure EV3. The statistics of mutations in the S protein of Omicron sub-variants.**

Comparing with the S protein of SARS-CoV-2 PT isolate, the mutation sites (including substitutions, deletions and insertions) in the S protein of Omicron sub-variants are highlighted in gold. The third type of amino acid on the same locus is marked in red and the fourth is labeled in blue. Deletion mutations are shown as blank squares.

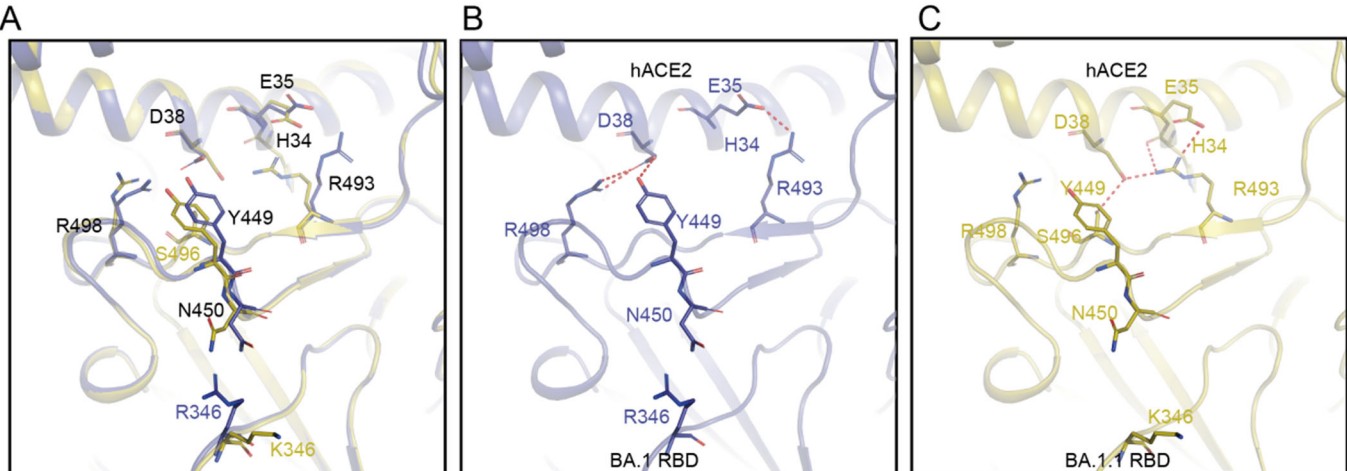

**Figure EV4.  The structural comparation of BA.1 and BA.1.1 RBDs in complex with hACE2.**

(A) The structural alignment of BA.1 and BA.1.1 RBDs bound to hACE2. (B, C) The interaction network of BA.1 RBD/hACE2 complex (B) and BA.1.1 RBD/hACE2 complex (C). The backbone of the structures are show as cartoon and the key residues are shown as sticks. The H-bonds and salt bridges are shown as red dash lines.

