## [Peer Review File · The EMBO Journal]

Key mechanistic features of the trade-off between antibody escape and host cell binding in the SARS-CoV-2 Omicron variant spike proteins

Weiwei Li, Zepeng Xu, Tianhui Niu, Yufeng Xie, Zhennan Zhao, Dedong Li, Qingwen He, Wenqiao Sun, Kaiyuan Shi, Wenjing Guo, Zhen Chang, Kefang Liu, Zheng Fan, Jianxun Qi, and George Gao

Corresponding authors: George Gao (gaof@im.ac.cn) , Zheng Fan (fanzh@im.ac.cn), Jianxun Qi (jxqi@im.ac.cn)

Review Timeline:

Submission Date:	21st Oct 23
Editorial Decision:	29th Nov 23
Revision Received:	8th Jan 24
Editorial Decision:	30th Jan 24
Revision Received:	7th Feb 24
Accepted:	9th Feb 24

Editor: Ieva Gailite

Transaction Report:

Dear Dr. Fan,

Thank you for submitting your manuscript to The EMBO Journal. I apologise for the protracted review process due to delays in referee report submission. We have now received comments from two reviewers, which are included below for your information.

As you will see from the reports, both reviewers find the study per se of interest, while also pointing out a number of reasonable concerns that would need to be addressed in the final version before they can recommend acceptance of the manuscript. Based on the interest expressed in the reports, I would like to invite you to address the issues raised by the referees in a revised manuscript. I should add that it is The EMBO Journal policy to allow only a single major round of revision and that it is therefore important to resolve the main concerns at this stage. I think it would be useful to discuss the revision in more detail via email or phone/videoconferencing - please let me know which option you prefer.

We generally allow three months as standard revision time, which can be extended to six months in the case of major revisions. As a matter of policy, competing manuscripts published during this period will not negatively impact on our assessment of the conceptual advance presented by your study. However, please contact me as soon as possible upon publication of any related work to discuss the appropriate course of action. Should you foresee a problem in meeting this deadline, please let us know in advance to discuss an extension.

When preparing your letter of response to the referees' comments, please bear in mind that this will form part of the Review Process File and will therefore be available online to the community. For more details on our Transparent Editorial Process, please visit our website: <https://www.embopress.org/page/journal/14602075/authorguide#transparentprocess>. Please also see the attached instructions for further guidelines on preparation of the revised manuscript.

Please feel free to contact me if have any further questions regarding the revision. Thank you for the opportunity to consider your work for publication, and I look forward to receiving the revised manuscript.

Yours sincerely,

Ieva Gailite

- a point-by-point response to the referees' comments, with a detailed description of the changes made (as a word file).
- a word file of the manuscript text.
- individual production quality figure files (one file per figure)
- a complete author checklist, which you can download from our author guidelines (<https://www.embopress.org/page/journal/14602075/authorguide>).

- Expanded View files (replacing Supplementary Information)

We realize that it is difficult to revise to a specific deadline. In the interest of protecting the conceptual advance provided by the work, we recommend a revision within 3 months (27th Feb 2024). Please discuss the revision progress ahead of this time with the editor if you require more time to complete the revisions.

Referee #1:

Summary:

The authors of this manuscript have shown the impacts from Omicron hotspot mutations (residue 346, 486, and 493) on the interaction between monomeric spike RBD and hACE2 receptor using structural methods including cryo-EM, X-ray crystallography and surface plasmon resonance (SPR). In comparison with their impacts on some RBD-targeting mAbs and spike-displaying VSV pseudovirus entry assay, the authors examine the compromise or balance between optimal receptor binding and immune evasion leading them to suggest a "two steps forward, one step back" model.

Overall, the cryo-EM and X-ray crystallography data have revealed detailed changes in selected sub-variant RBDs' interactions with ACE2.

Overall the manuscript is well written and clear.

General comments:

1. There is a clear gap drawing conclusion from the results of RBD-hACE2 binding affinity and Spike-hACE2 initiated viral entry. Using the monomeric RBD for SPR binding studies provides quantitative assessment of monovalent RBD-ACE2 interactions most closely resembling RBD in open conformation, however this neglects numerous other factors that are present when intact trimeric spikes and native ACE2 receptors are interacting. Such other factors include RBD open/closed conformations, NTD influence on RBD accessibility may also cause apparent binding affinity between spike and hACE2 trend departing from the results of RBD-hACE2 interaction, and native ACE2 is dimeric. Thus the RBD-hACE2 SPR data, while informative, are not a direct reflection of how a given sub-variant's spike will engage with receptor.
2. Similarly, the S copy number on the surface of authentic virus or pseudovirus may change with sub-variant and as a result of mutations, and these changes could alter avidity for receptor and antibodies. It is important to attempt to take these factors into account.
3. For the pair-wise comparison between BA.4/5 & BF.7, BQ.1 & BQ.1.1 and XBB & XBB.1.5, it is more informative to also include k-on and k-off in the SPR results. That should also reinforce the cryo-EM data on the binding kinetics and modes between RBD variants and hACE2.
4. When "It is R493 but not Q493 that is regulated by residue 346 in the RBD" and the later "two steps forward, one step back" were discussed based on the RBD-hACE2 binding affinity upon mutations at 346, 486 and 493, is it more direct and informative to show another comparison between BA.2.75 and BM.1.1.1 or XBB where two mutations occurred at the same time.
5. The choice of mAbs raises some questions. Is it possible to draw the conclusions they attempt to that link immune evasion and receptor affinity, would seem more appropriate to use polyclonal sera or RBD mAbs from patients infected with those specific variants and sub-variants. At the least a more thorough explanation and rationale arguing for the appropriateness of the choice of mAbs is needed.

Referee #2:

Since its emergence in late 2021 the SARS-CoV-2 omicron variant has evolved into several sublineages that took over dominance over the COVID-19 pandemic. During the course of their emergence and diversification, several omicron sublineages developed additional mutations in the receptor binding domain (RBD) of the spike protein that improved ACE2 binding and extended their capacity to evade neutralization by antibodies.

In this study, Li and colleagues crystallized ACE2 complexes with RBDs from various omicron sublineages. They identified interacting amino acid residues and evaluated the impact of sublineage-specific RBD mutations on ACE2 binding efficacy and antibody evasion through SPR-based protein binding studies. The findings revealed that in BA.5 sublineages, the presence of the RBD mutation R346T, which is crucial for antibody evasion, regulates the RBD residue R493 for optimal RBD/ACE2 interaction. Conversely, in BA.2 sublineages, lacking RBD mutation R346T but featuring RBD mutation F486V as a key mediator of antibody evasion, RBD residue Q493 is favored, despite causing a moderate reduction in immune escape. This balancing effect between mutations R493Q and F486V achieves strong, albeit sub-optimal, immune evasion while maintaining efficient ACE2/RBD interaction.

This study, characterized by its high quality, offers valuable insights into the evolution of omicron sublineages and the functional consequences of combined RBD mutations. The manuscript is well-written, and the data are clearly presented. I recommend accepting this manuscript after addressing minor modifications.

Minor points:

- While RBD mutations directly impact ACE2/RBD interaction, it's worth discussing that mutations outside the RBD may affect ACE2 binding by influencing RBD exposure in the "up" conformation. The limitations of working exclusively with isolated RBDs should also be acknowledged in this context.
- Lines 372-374, please provide detailed description (including primer sequences) on how pseudoviruses were quantified.
- Figure 2B (patch 1), please add label for residue Y83
- Line 144, "shorter" instead of "short"
- Line 262, "RBD"
- Lines 264, 265, "MAbs were" instead of "MAbs was"

Point-by-point letter:**Referee #1:**

Summary:

The authors of this manuscript have shown the impacts from Omicron hotspot mutations (residue 346, 486, and 493) on the interaction between monomeric spike RBD and hACE2 receptor using structural methods including cryo-EM, X-ray crystallography and surface plasmon resonance (SPR). In comparison with their impacts on some RBD-targeting mAbs and spike-displaying VSV pseudovirus entry assay, the authors examine the compromise or balance between optimal receptor binding and immune evasion leading them to suggest a "two steps forward, one step back" model.

Overall, the cryo-EM and X-ray crystallography data have revealed detailed changes in selected sub-variant RBDs' interactions with ACE2.

Overall the manuscript is well written and clear.

Response: Thank you for these positive comments.

General comments:

1. There is a clear gap drawing conclusion from the results of RBD-hACE2 binding affinity and Spike-hACE2 initiated viral entry. Using the monomeric RBD for SPR binding studies provides quantitative assessment of monovalent RBD-ACE2 interactions most closely resembling RBD in open conformation, however this neglects numerous other factors that are present when intact trimeric spikes and native ACE2 receptors are interacting. Such other factors include RBD open/closed conformations, NTD influence on RBD accessibility may also cause apparent binding affinity between spike and hACE2 trend departing from the results of RBD-hACE2 interaction, and native ACE2 is dimeric. Thus the RBD-hACE2 SPR data, while informative, are not a direct reflection of how a given sub-variant's spike will engage with receptor.

Response: We agree with the reviewer's comment that other factors, such as RBD open/closed conformations and NTD influence on RBD accessibility, may also contribute to the apparent binding affinity between spike and hACE2. In response to this feedback, we

conducted additional experiments to evaluate the binding affinities of spike (S) proteins to hACE2. We found that the changing trends of RBD-hACE2 affinities and S-ACE2 affinities of these sub-variants were consistent. However, it is worth noting that since both S and RBD are immobilized proteins, our findings may not fully reflect the effect of RBD open/closed conformations on affinity. This limitation is inherent to the SPR experimental methodology. We added this limitation in the last paragraph of Discussion.

2. Similarly, the S copy number on the surface of authentic virus or pseudovirus may change with sub-variant and as a result of mutations, and these changes could alter avidity for receptor and antibodies. It is important to attempt to take these factors into account.

Response: Thank you for your comments. We appreciate your input regarding the limitations of our experiments, and we added this limitation to the Discussion in the revision manuscript.

3. For the pair-wise comparison between BA.4/5 & BF.7, BQ.1 & BQ.1.1 and XBB & XBB.1.5, it is more informative to also include k-on and k-off in the SPR results. That should also reinforce the cryo-EM data on the binding kinetics and modes between RBD variants and hACE2.

Response: The k-on and k-off of the SPR was provided in the Table S1 and S5 in the revised manuscript.

4. When "It is R493 but not Q493 that is regulated by residue 346 in the RBD" and the later "two steps forward, one step back" were discussed based on the RBD-hACE2 binding affinity upon mutations at 346, 486 and 493, is it more direct and informative to show another comparison between BA.2.75 and BM.1.1.1 or XBB where two mutations occurred at the same time.

Response: This is a good suggestion and we have carried out the new experiments with new data to support this notion. The RBD of BM.1.1.1 has three substitutions (R346T, F486S, and F490S) compared to the BA.2.75 RBD. The binding affinity of BM.1.1.1 RBD to hACE2 ($K_D = 71.70 \pm 1.71$ nM) is lower than that of BA.2.75 ($K_D = 8.21 \pm 0.59$ nM),

due to the F486S substitution. When Q493 was substituted with R493, the binding affinity of BA.2.75 RBD Q493R to hACE2 decreased ($K_D = 31.47 \pm 0.26$ nM), while the binding affinity of BM.1.1.1 RBD Q493R increased ($K_D = 28.37 \pm 0.39$ nM). These changes are mainly due to the difference in residue 346 between the BM.1.1.1 and BA.2.75 RBDs.

5. The choice of mAbs raises some questions. Is it possible to draw the conclusions they attempt to that link immune evasion and receptor affinity, would seem more appropriate to use polyclonal sera or RBD mAbs from patients infected with those specific variants and sub-variants. At the least a more thorough explanation and rationale arguing for the appropriateness of the choice of mAbs is needed.

Response: Q493 is a reverse mutation that affects the binding epitope of RBD-1 antibodies. Prior to the emergence of Omicron BA.1, the residue 493 in the RBD was Q. However, after the emergence of Omicron BA.1, Q493 was replaced by R493. Subsequently, when SARS-CoV-2 evolved from BA.2 to BA.4/5, R493 mutated back to Q493. To evaluate whether antibodies induced by sub-variants prior to Omicron BA.1 reverse some ability to neutralize the SARS-CoV-2 Omicron sub-variants after the Q493 reverse mutation, we selected these antibodies for testing. This has been described in the revised manuscript on lines 212-214.

Referee #2:

Since its emergence in late 2021 the SARS-CoV-2 omicron variant has evolved into several sublineages that took over dominance over the COVI19 pandemic. During the course of their emergence and diversification, several omicron sublineages developed additional mutations in the receptor binding domain (RBD) of the spike protein that improved ACE2 binding and extended their capacity to evade neutralization by antibodies.

In this study, Li and colleagues crystallized ACE2 complexes with RBDs from various omicron sublineages. They identified interacting amino acid residues and evaluated the impact of sublineage-specific RBD mutations on ACE2 binding efficacy and antibody evasion through SPR-based protein binding studies. The findings revealed that in BA.5 sublineages,

the presence of the RBD mutation R346T, which is crucial for antibody evasion, regulates the RBD residue R493 for optimal RBD/ACE2 interaction. Conversely, in BA.2 sublineages, lacking RBD mutation R346T but featuring RBD mutation F486V as a key mediator of antibody evasion, RBD residue Q493 is favored, despite causing a moderate reduction in immune escape. This balancing effect between mutations R493Q and F486V achieves strong, albeit sub-optimal, immune evasion while maintaining efficient ACE2/RBD interaction.

This study, characterized by its high quality, offers valuable insights into the evolution of omicron sublineages and the functional consequences of combined RBD mutations. The manuscript is well-written, and the data are clearly presented. I recommend accepting this manuscript after addressing minor modifications.

Response: Thank you for these positive comments.

Minor points:

- While RBD mutations directly impact ACE2/RBD interaction, it's worth discussing that mutations outside the RBD may affect ACE2 binding by influencing RBD exposure in the "up" conformation. The limitations of working exclusively with isolated RBDs should also be acknowledged in this context.

Response: Thank you for the suggestion. We have acknowledged this limitation in the Discussion.

- Lines 372-374, please provide detailed description (including primer sequences) on how pseudoviruses were quantified.

Response: We described the pseudoviruses quantification in detail and provided the primer sequences in the revised manuscript.

- Figure 2B (patch 1), please add label for residue Y83

Response: Labeled.

- Line 144, "shorter" instead of "short"

Response: Corrected.

- Line 262, "RBD"

Response: Corrected.

- Lines 264, 265, "MAbs were" instead of "MAbs was"

Response: Corrected.

Dear George,

Thank you for submitting a revised version of your manuscript. Your study has now been seen by one of the original referees, who finds that their previous concerns have been addressed and now recommend acceptance of the manuscript.

There now remain only a few editorial points that need addressing before I can extend acceptance of the manuscript:

1. Email to the co-corresponding author Zheng Fan did not reach the addressee - please double check the provided address (fanz@im.ac.cn).
2. Please make sure that the order of the sections in the manuscript is as follows: abstract, introduction, results, discussion, materials & methods, data availability section, acknowledgments, disclosure statement and competing interests, references, main figure legends, tables, expanded figure legends.
3. Please remove table legends from the manuscript text file.
4. We are missing the ORCID iD for the co-corresponding author Zheng Fan. In order to link the ORCID iD to the account in our manuscript tracking system, the author in question has to do the following:
 - Click the 'Modify Profile' link at the bottom of your homepage in our system.
 - On the next page you will see a box halfway down the page titled ORCID*. Below this box is red text reading 'To Register/Link to ORCID, click here'. Please follow that link: you will be taken to ORCID where you can log in to your account (or create an account if you don't have one)
 - You will then be asked to authorise Wiley to access your ORCID information. Once you have approved the linking, you will be brought back to our manuscript system.Unfortunately, we cannot do this linking on the author's behalf for security reasons.
5. We can accommodate up to five EV figures. Please move the rest of the EV figures and their legends to the Appendix, which should be prefaced with a brief table of contents. Further information is available here: <https://www.embopress.org/page/journal/14602075/authorguide#expandedview>.
6. CRediT has replaced the traditional author contributions section because it offers a systematic, machine-readable author contributions format that allows for more effective research assessment. Please remove the Authors Contributions from the manuscript and use the free text boxes beneath each contributing author's name in our online submission system to add specific details on the author's contribution. More information is available in our guide to authors.
7. Please rename "Conflict of interest" section into "Disclosure and competing interests statement" (further info: <https://www.embopress.org/page/journal/14602075/authorguide#conflictsofinterest>).
8. According to our updated Author guidelines (<https://www.embopress.org/competing-interests?=#disclosure-statement>), please state in the "Disclosure Statement & Competing Interests" section that Dr. Gao is a member of the editorial advisory board at The EMBO Journal. Our recommended formulation is as follows:
"George Gao is an editorial advisory board member at The EMBO Journal. This has no bearing on the editorial consideration of this article for publication."
9. In the Data Availability section, please add resolvable links to the datasets. More information about the format of this section can be found here: <https://www.embopress.org/page/journal/14602075/authorguide#dataavailability>.
10. Our data editors have indicated that the measure of the centre for the error bar needs to be defined in the legend of figure 1a.
11. Papers published in The EMBO Journal are accompanied online by a 'Synopsis' to enhance discoverability of the manuscript. It consists of A) a short (1-2 sentences) summary of the findings and their significance, B) 3-4 bullet points highlighting key results and C) a synopsis image that is 550x300-600 pixels large (width x height, jpeg or png format). You can either show a model or key data in the synopsis image. Please note that the image size is rather small and that text needs to be readable at the final size. Please send us this information together with the revised manuscript.

With best wishes,

Ieva

Ieva Gailite, PhD
Senior Scientific Editor
The EMBO Journal
Meyerhofstrasse 1
D-69117 Heidelberg

Tel: +4962218891309
i.gailite@embojournal.org

Further information is available in our Guide For Authors: <https://www.embopress.org/page/journal/14602075/authorguide> We realize that it is difficult to revise to a specific deadline. In the interest of protecting the conceptual advance provided by the work, we recommend a revision within 3 months (29th Apr 2024). Please discuss the revision progress ahead of this time with the editor if you require more time to complete the revisions.

Referee #1:

We appreciate the additional experiments and data provided by the authors. Also the updated discussion about limitations and caveats is helpful.

The authors addressed the editorial issues.

Dear George,

Thank you for addressing the final editorial issues. I am now pleased to inform you that your manuscript has been accepted for publication.

I will look into the synopsis text in the next couple of days and let you know if any edits to the journal style are needed.

Finally, we would like to promote your manuscript among the Chinese readership. Therefore, we would like to invite you to prepare a short summary of the manuscript in Chinese (1500-2000 Chinese characters), which we will promote on the WeChat platform 'BioArt' with more than 610,000 followers.

If you are interested in this opportunity, we recommend covering the article very close to its online publication date. Thus, ideally we would very much appreciate if you could send us a draft within the next 7 working days. Please let us know whether or not you would be interested in contributing such a short summary in Chinese.

I have included below some general guidelines on how to prepare a summary and a link to recent examples for your reference. Please let me know if you have any questions about this.

If you have any questions, please do not hesitate to contact the Editorial Office. Thank you for this contribution to The EMBO Journal and congratulations on a great paper!

Best wishes,

Ieva

General WeChat Summary Guidelines

1. These summary articles are meant to be targeting general audience so please limit the use of specialized technical terms, acronyms and jargon.
2. A summary usually starts with brief background information of the reported work, which is followed by explaining the findings in some detail, and ends with a short review of the conclusions as well as the implications of the work and future directions for the research.
3. The summary should at least contain one graphical item, such as a scheme or a figure from the paper.
4. Please provide ONE SINGLE document containing all text and graphical materials, ideally as a Word.docx or .doc file. Please DO NOT provide the document as a .pdf file.
5. Please DO NOT publicly release the document before the paper is officially published online.

Summary Examples

EMBO J | 罗招庆/欧阳松应揭示谷酰胺脱氨酶MvcA的去泛素化功能

EMBO J | 王松灵院士团队揭示组织内应力调控大型哺乳动物乳恒牙替换的新机制

*